# Assessing Reclaimed Urban Wastewater for Reuse in Agriculture: Technical and Economic Concerns for Mediterranean Regions

**Giacomo Giannoccaro** [1],*, **Stefania Arborea** [2], **Bernardo C. de Gennaro** [1], **Vito Iacobellis** [2] **and A. Ferruccio Piccinni** [2]

1    University of Bari, Department DiSAAT, Via Amendola 165/A, 70126 Bari, Italy
2    Politecnico di Bari, Department DICATECh, Via Orabona 4, 70126 Bari, Italy
*    Correspondence: giacomo.giannoccaro@uniba.it; Tel.: +39-08-0544-2885

**Abstract:** Direct reuse of treated wastewater can offer a realistic supply alternative for irrigation in Mediterranean areas. In this study, we conducted a spatial cost-benefit analysis to quantify and locate the volume of technically and economically feasible and readily available reclaimed urban wastewater. We considered the case of Puglia (Italy) and the results are discussed in terms of the implications for policy-making and pointing out future research needs. The results showed that the main technical barrier is the shortness of the irrigation season. On the other hand, the main economic concern is related to filtration followed by lack of conveyance systems. While our results are based on estimates, future research should try to include practical experiments based on actual data. Further research should also address the issue of transaction costs by establishing the obligations of wastewater treatment plants to deliver reclaimed water to farmers.

**Keywords:** wastewater treatment; reuse; irrigation; groundwater; spatial cost-benefit analysis

---

## 1. Introduction

The general decreasing trend in water availability and the need for sustainable use of available water resources have led regional and national governments worldwide to seek alternative water sources. Desalination and wastewater treatment are thought to have the greatest potential. According to the European Wastewater Directive [1], all wastewater must be treated before it can be disposed of in natural water bodies. However, before treated wastewater can be directly used in agriculture, it requires an additional (tertiary) treatment to convert it into reclaimed wastewater.

As agriculture is the largest water user in many Mediterranean regions, there has long been a consensus that the direct reuse of reclaimed wastewater can offer a realistic supply alternative for irrigation [2–5]. In addition to increasing water availability, wastewater treatment is also expected to benefit the environment, as recognised by the European Water Framework Directive [6]. While in developing countries the main advantage of improved treatment is that it reduces the amount of pollutants released into surface water bodies [7], in developed areas refined wastewater is of particular importance for maintaining minimum flow levels of river systems [8], restoring existing wetlands [9] and creating new wetlands and recreational areas [10].

Most areas along Mediterranean coasts are experiencing the detrimental effects of seawater intrusion as a consequence of over-exploitation of groundwater. In this regard, reclaimed wastewater can be used either to reduce the pumping rate of groundwater as a complementary irrigation source [5,11] or, where envisaged under national law, as an artificial aquifer to replenish groundwater [12,13].

The use of reclaimed wastewater for irrigation can positively affect plant growth by providing supplementary nutrients to crops and, with appropriate crop management, considerably reduce

the use of fertilisers [14,15]. Alternatively, from a circular economy perspective, the recovery of phosphorus from wastewater treatment plants (WWTP) as a substitute for chemical P fertilisers has been proposed [16].

While the feasibility of wastewater recycling has long been demonstrated, technological advancements are also emerging [17]. Although technological progress ensures that recycling is safe [18,19], the total volume of treated wastewater reused in Europe only represents a very small percentage of the treated effluent. Water reuse projects may fail for various reasons. One is the lack of popular support, because the perceived risk of poor water quality leads to problems with acceptance [20,21]. In developing countries the local capacity to utilise suitable technologies is a crucial problem, as treated wastewater can also be a source of pathogenic organisms and potentially hazardous chemical substances [7]. On top of these issues, the main factor hampering the development of WWTPs for reuse is related to the total costs of reclamation (plant construction, operation and maintenance), of distribution and of monitoring the reuse system as a whole [22].

Numerous studies have investigated the feasibility of WWTPs for making reclaimed wastewater reusable in agriculture. While the technological and agronomic aspects of wastewater reuse in agriculture have been widely investigated, the economic feasibility of WWTPs for reuse has, as a whole, been studied less [11]. The common approach to assess the economic feasibility of WWTPs for reuse is cost-benefit analysis (CBA) [23], which compares all of the costs incurred and benefits achieved from reclaimed wastewater reuse. Nevertheless, in the case of WWTPs for reuse, economic estimates are not straightforward. While there could be some context-specific issues, the main difficulty regards the commensurability of (i) the place where and (ii) the time when the costs and benefits occur. The most relevant aspect is probably the place where the costs and benefits are measured, namely on-site versus at source [11]. As far as time is concerned, the short-term nature of cost incidence diverges from the long-term nature of benefit attainment, particularly in terms of the environment. In this regard, the discount rate makes the difference, as discussed in [13].

In this context, the aim of this study is to improve the assessment of reclaimed wastewater for reuse in agriculture. In particular, it attempts to shed light on the technical and economic challenges for Mediterranean areas. We used the region of Puglia (in south-eastern Italy) as a case study to assess reclaimed urban wastewater for irrigation. We conducted a spatial cost-benefit analysis in order to quantify and locate the volume of technically and economically feasible and readily available reclaimed urban wastewater. The results of the cost-benefit analysis are discussed, considering their implications for policy-making and pointing out future research needs.

## 2. Materials and Methods

### 2.1. Framework Analysis

The spatial cost-benefit analysis was carried out in a sequence of four main cascaded steps. Firstly, the technically feasible volume was determined; this refers to the potential raw volume of reclaimed urban wastewater based on the number of treatment plants and their capacity in terms of population equivalent (PE) and, on the irrigation water demand. Secondly, the economic feasibility of reclamation treatments (i.e., tertiary) for reuse in agriculture was defined. Thirdly, the volume of readily available reclaimed wastewater was evaluated against the existing irrigation networks. Increasing reuse of reclaimed wastewater can replace groundwater resources and consequently alleviate over-exploitation. Thus, the reclaimed wastewater used for irrigation was spatially located using the vulnerable groundwater map of Puglia.

The resulting data were used to assess the volume of economically feasible and readily available reclaimed urban wastewater within the groundwater vulnerable zone.

Chart flow of methodological steps is shown in Figure 1.

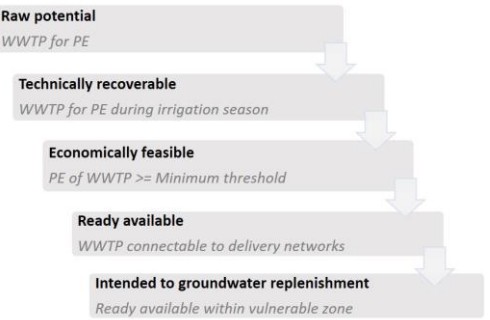

**Figure 1.** Methodological steps.

### 2.1.1. Recoverable Reclaimed Urban Wastewater

Recoverable urban wastewater refers to the technically feasible volume. It depends on the number of plants and, consequently, on PE distribution in Puglia as well as on the irrigation water demand. Indeed, reclaimed wastewater can be valuable as long as irrigation water demand rises up. The PE is the unit applied to wastewater to describe the size of package sewage treatment plants. Moreover, at any time each plant may or may not be provided with tertiary treatment processes or be at a different stage of work (i.e., already operating, constructed, to be upgraded or to be built).

The regional government's wastewater treatment plan [24] recognises 93 treatment plants for making wastewater reusable after suitable reclamation treatments. Some of these are already equipped with tertiary treatment systems while others are not. On the other hand, there are a number of plants that are already equipped with tertiary systems but are in need of upgrading because they are obsolete or have broken down.

While the potential volume of reclaimed wastewater is based on plant size (PE) (an example of a calculation is provided in the Supplementary material of [11]), the demand for irrigation water basically depends on the climate. The case study refers to a region that has a Mediterranean climate, characterised by warm-to-hot dry summers and mild-to-cool wet winters. Irrigation is important for the overall economy of the region, especially for agriculture. Demand for irrigation water is highest in spring and summer and the study considered a six-month period of demand for irrigation water. (As a reviewer pointed out reclaimed wastewater could be used for aquifer recharge the rest of the time. Although such use could still be beneficial given the general situation of groundwater stress. It should be noticed that, within Italian regulation, artificial groundwater recharge is forbidden).

### 2.1.2. Economic Feasibility of Reclaimed Urban Wastewater for Reuse in Agriculture

The methodological framework for assessing the economic feasibility of reclaimed wastewater for reuse in agriculture in Puglia was developed by Arborea et al. [11]. Within this framework, only the real economic benefits of reclaimed water as a productive factor for irrigation are taken into account while environmental benefits are not. Additional reclamation costs to make treated wastewater reusable are analysed in relation to the current effluent discharge of WWTPs (Italian laws-D.L.152/06 [25] and D.M.185/2003 [26], establish the effluent standard quality requirements for surface water discharge and ground surface discharge (Appendix A)). Furthermore, in the case of plants with current effluent discharge into surface water, the effect of the temperature (at 15 °C and 20 °C) of primary sedimentation for nitrification and denitrification processes is analysed. Namely, an average of 15 °C for plants that work throughout the year and 20 °C for plants that operate seasonally in spring and summer, during the irrigation period in Mediterranean areas.

In principle, the temperature affects: (i) the relationship between the volumes of nitrogen and biodegradable organic matter removed; (ii) increases in volume of biodegradable organic matter removed and (iii) the relationship between nitrification and denitrification volumes. Hence, the temperature directly affects the dimensions (volume and surface area) of the nitrification and denitrification tanks, and, as a consequence, the related construction costs. The temperature is



also a factor that affects the costs of electromechanical systems and energy for aeration of the mixture to be treated.

According to the Italian regulation (Appendix A), when upgrading from plants discharging into surface water, filtration is always considered necessary for reaching the standards of reuse; hence we have always considered filtration as a required treatment for reuse (Yes in Table 1). On the other hand, filtration is considered an enhanced treatment for reaching standards of reuse when upgrading plants that discharge onto the ground. In this case filtration is often already present in WWTPs in Puglia. Then, for the cost assessment we have to consider two cases: (i) WWTP is not provided with filtration, therefore it has to be considered in the costs for upgrading (Yes in Table 1), (ii) WWTP already relies on filtration, therefore it is not considered in the costs for upgrading (No in Table 1). Both cases are considered technically possible in our general analysis.

**Table 1.** Minimum size (PE) of WWTPs for economic feasibility of tertiary treatment.

| Wastewater Use | Current Effluent Discharge | Primary Temperature | Filtration | Minimum PE Threshold |
|---|---|---|---|---|
| New irrigated land | Surface water | 15 °C | Yes | none |
| | | 20 °C | Yes | 175,000 |
| | Ground | 15 °C | Yes | 100,000 |
| | | | No | 5000 |
| Preserving groundwater from salt intrusion | Surface water | 15 °C | Yes | 250,000 |
| | | 20 °C | Yes | 100,000 |
| | Ground | 15 °C | Yes | 30,000 |
| | | | No | 5000 |

Note: The irrigation water value is 0.21 EUR/m$^3$ from [26]; the economic benefit of preserving groundwater is 0.22 EUR/m$^3$ from [27]. Source: adapted from [11].

We evaluated the investments necessary to implement filtration treatment, we considered two commonly exploited technologies: gravity filtration or pressure units and evaluated costs with reference to the plant potential in terms of the PE. The estimated costs of upgrades showed interesting scale effects. In particular, pressure filters are advantageous for installations ranging from 2000 to 100,000 PE while gravity filters are convenient for higher PEs. As filtration is the main item for economic feasibility, the reader can find many more details on the cost in [11] with the related supplementary materials.

Furthermore, our study considers two main hypotheses, namely: (i) reclaimed wastewater used for newly irrigated land (Hypothesis I); and (ii) reclaimed wastewater as complementary to current groundwater sources (Hypothesis II). The first hypothesis considers the simplest case, in which rainfed farmland is supplied with reclaimed wastewater. In this case, the costs at plant gate would include additional treatment costs and the benefits are those associated with the direct use of reclaimed urban wastewater for irrigation. The irrigation water value is from [27], taking the region's average value of 2475 m$^3$/ha as the irrigation volume. The second hypothesis reflects the regional government's aim to replace groundwater sources with reclaimed urban wastewater. In this case, the benefit of groundwater replenishment is also accounted for, keeping invariant additional reclamation costs. The economic benefit of preserving groundwater from salt intrusion in Puglia is reported in [28].

Finally, PE capacity is used to determine the size at which, according to the treatment features and taking the irrigation water value of 0.21 EUR/m$^3$ as assessed in [27] or the economic benefit of preserving groundwater from salt intrusion (0.22 EUR/m$^3$) as reported in [28], it is economically feasible to provide WWTPs with tertiary treatment for irrigation purposes (Table 1).

In general, the reclamation costs are only sustainable for medium to large wastewater treatment plants ($\geq$5000 PE), in line with [22].

The third step of the methodology deals with the cost of making reclaimed urban wastewater available at the right place (i.e., the farm gate). Thus, the characteristics of wastewater treatment plants were integrated, through the GIS, with data on collective delivery networks for irrigation [22]. In Puglia,

the average size of the irrigated land is smaller than 5 ha with 63,909 farms using irrigation (23.5% of total farms) [29]. In addition, half of the farms that use irrigation have on-farm wells, while almost 60% of irrigation water is derived from groundwater resources [30]. In Puglia, there are six Reclamation and Irrigation Boards (RIB): in Gargano, Capitanata, Arneo, Stornara e Tara, Terre d'Apulia, and Ugento Lì Foggi (Table 2). The situation differs across the region. For example, while the Province of Foggia represents the best example of a collective irrigation delivery system for surface water, within the Province of Lecce, almost 80% of the irrigation water is derived from direct on-farm access to groundwater resources. In addition, considering the lack of surface water bodies, collective irrigation delivery systems in the Province of Lecce mostly rely on groundwater sources.

**Table 2.** Main features of collective delivery networks for irrigation water in Puglia.

| Reclamation Consortia | Total Area | Equipped Area | Operating Area |
|---|---|---|---|
| | **Thousands Hectares** | | |
| Gargano | 150.337 | 0.975 | 0.946 |
| Capitanata | 441.545 | 147.131 | 147.131 |
| Terre D'Apulia | 569.807 | 28.086 | 22.878 |
| Stornara e Tara | 142.949 | 42.042 | 22.934 |
| Arneo | 252.981 | 16.252 | 4.860 |
| Ugento Li Foggi | 189.494 | 10.775 | 10.775 |
| Total | 1747.113 | 245.261 | 209.524 |

Source: [24].

The last step of the methodology consists of a spatial distribution of readily available reclaimed wastewater as previously assessed across the aquifers shown on the map of Puglia. The groundwater vulnerable zone is shown in Figure 2.

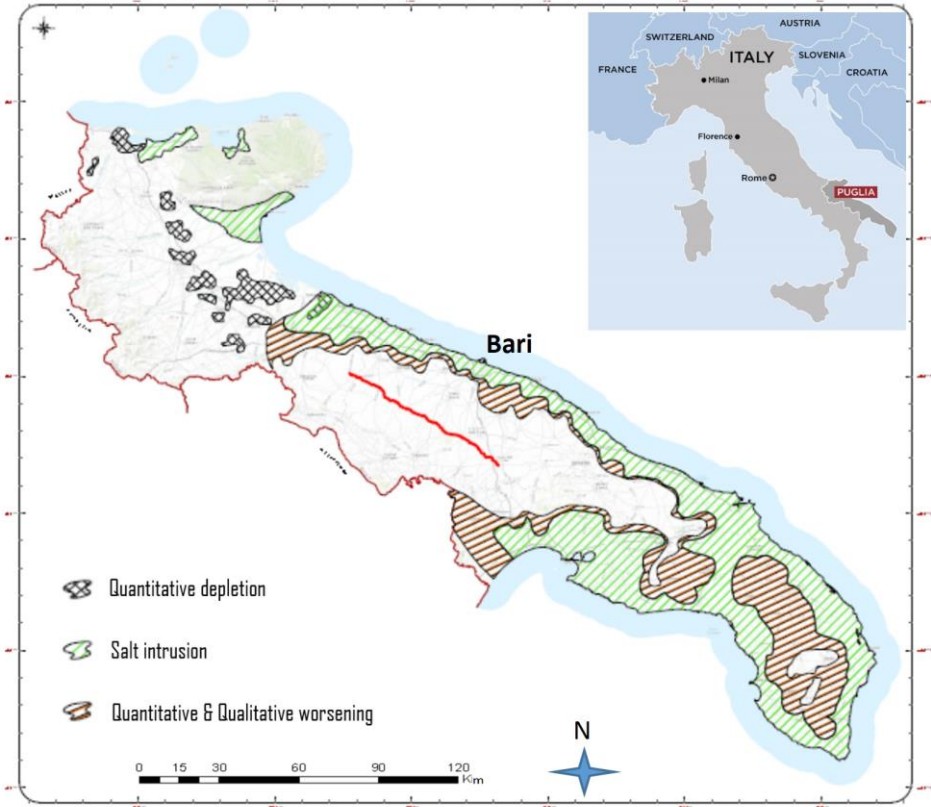

**Figure 2.** Groundwater vulnerable zone in Puglia [31].

It is worth mentioning that Hypothesis II is only applicable across the vulnerable zone. The aim of this last step is to demonstrate the reliability of reclaimed urban wastewater for reuse in agriculture as a solution to groundwater over-abstraction.

## 3. Results

### 3.1. Reclaimed Urban Wastewater: Technically Recoverable for Irrigation

First of all, the number of WWTPs: (a) operating, (b) constructed (and ready to work), (c) existing but will be upgraded and (d) to be built, in Puglia in 2015 is reported in Table 3.

**Table 3.** Number of plants according to wastewater treatment planning (2015).

| Province | Operating | Constructed | To Be Upgraded | To Be Built | Total |
|---|---|---|---|---|---|
| Bari | | 3 | 5 | 13 | 21 |
| Barletta-Trani-Andria | | 2 | 3 | | 5 |
| Brindisi | 2 | | 8 | 4 | 14 |
| Foggia | | 2 | 6 | 3 | 11 |
| Lecce | 2 | 2 | 4 | 18 | 26 |
| Taranto | | 2 | 7 | 7 | 16 |
| **Total** | **4** | **11** | **33** | **45** | **93** |

Source: adapted from [24].

The number of WWTPs grouped by plant size (PE), according to the stage of operation and their current effluent discharge is shown in Table 4.

**Table 4.** Classification of wastewater plants in Puglia based on current effluent discharge and plant size (2015).

| Plant Size (PE) | Operating Surface Water | Operating Ground | Constructed * Surface Water | Constructed * Ground | To Be Upgraded Surface Water | To Be Upgraded Ground | To Be Built Surface Water | To Be Built Ground | Total Surface Water | Total Ground |
|---|---|---|---|---|---|---|---|---|---|---|
| 5000 | | | | | 1 | 2 | | | 1 | 2 |
| 10,000 | | | | 1 | 1 | 1 | | 2 | 1 | 4 |
| 20,000 | | 1 | 2 | 2 | 3 | 6 | 1 | 14 | 6 | 23 |
| 30,000 | | | | 1 | 1 | 3 | 1 | 8 | 2 | 12 |
| 40,000 | 1 | | | 1 | | 2 | 1 | 6 | 2 | 9 |
| 50,000 | 1 | | | | | 1 | 1 | 5 | 2 | 6 |
| 70,000 | | | | 1 | 1 | 2 | | 1 | 1 | 4 |
| 100,000 | 1 | | 1 | 1 | 2 | | 3 | 2 | 7 | 3 |
| 250,000 | | | | | 5 | 1 | | | 5 | 1 |
| 500,000 | | | | | 1 | | | | 1 | |
| **Total** | **3** | **1** | **3** | **7** | **15** | **18** | **7** | **38** | **28** | **64** |
| | **4** | | **10** | | **33** | | **45** | | **92** | |

Source: adapted from [24]. * Data not available for WWTP of Uggiano la Chiesa.

Table 5 shows the technically recoverable annual volume for irrigation.

The total technically recoverable volume amounts to 96,834,087 m$^3$ per year, namely half of the potential volume as a consequence of the six-month period for irrigation. If all upgrading operations were completed on all types of plants (constructed, to be upgraded and, to be built), the recoverable volume of reclaimed wastewater would increase from less than 4 million m$^3$ per year (from those WWTPs with tertiary process already operating) to about 97 million m$^3$ per year. As reported in Table 5, the largest amount of reclaimed wastewater would come from WWTPs classified as *to be upgraded*, followed by those *to be built*. This would involve huge financial investments.

**Table 5.** Technically recoverable annual volumes (m$^3$) of reclaimed urban wastewater for irrigation in Puglia (2015).

| Plant Size (PE) | Operational Stage | | | | | | | |
|---|---|---|---|---|---|---|---|---|
| | Operating | | Constructed | | To Be Upgraded | | To Be Built | |
| | Current Effluent Quality | | | | | | | |
| | Surface Water | Ground | Surface Water | Ground | Surface Water | Ground | Surface Water | Ground |
| 5000 | | | | | | 210,240 | | |
| 10,000 | | | 219,00 | | 547,500 | 262,331 | | 618,811 |
| 20,000 | | 237,681 | 978,054 | 647,876 | 1,586,327 | 2,309,535 | 352,000 | 5,597,758 |
| 30,000 | | | | 448,096 | 822,695 | 1,487,960 | 657,000 | 4,907,615 |
| 40,000 | 946,080 | | | 912,499 | | 7,637,454 | 357,977 | 3,946,818 |
| 50,000 | 1,016,896 | | | | | 827,894 | 1,916,250 | 7,022,936 |
| 70,000 | | | 1,454,160 | | 683,280 | 1,867,085 | 394,000 | 613,813 |
| 100,000 | 1,620,600 | | 1,650,069 | 2,416,214 | 5,022,511 | | 3,401,403 | 4,355,990 |
| 250,000 | | | | | 8,479,356 | 2,522,823 | | |
| 500,000 | | | | | 15,877,500 | | | |
| Total | 3,583,576 | 237,681 | 2,628,123 | 6,097,845 | 33,019,169 | 17,125,322 | 7,078,630 | 27,063,741 |
| | 3,821,257 | | 8,725,968 | | 50,144,491 | | 34,142,371 | |

*3.2. Reclaimed Urban Wastewater: Economically Feasible and Readily Available*

In order to assess the economic feasibility and determine the total recoverable volume, the technical features of WWTPs as reported in Table 4 were crossed with the minimum PE threshold (Table 1) to find the number of economically sustainable plants. The results of this analysis are reported in Table 6.

**Table 6.** Economically feasible reclaimed urban wastewater volume in Puglia (2015).

| Wastewater Reuse | Current Effluent Discharge | Primary Temperature | Filtration | PE | Plants | Volume (m$^3$/Year) |
|---|---|---|---|---|---|---|
| New irrigated land | Surface water | 15 °C | Yes | None | 0 | 0 |
| | | 20 °C | Yes | 175,000 | 2 | 21,380,278 |
| | Ground | 15 °C | Yes | 100,000 | 1 | 2,522,823 |
| | | | No | 5000 | 63 | 48,030,563 |
| Preserving groundwater from salt intrusion | Surface water | 15 °C | Yes | 250,000 | 1 | 15,877,500 |
| | | 20 °C | Yes | 100,000 | 3 | 24,356,856 |
| | Ground | 15 °C | Yes | 30,000 | 25 | 33,018,174 |
| | | | No | 5000 | 63 | 48,030,563 |

In the case of irrigation of new land with reclaimed urban wastewater (Hypothesis I), only two plants of those discharging into surface water meet the minimum PE threshold, and just one plant of those discharging onto ground, when the filtration process is included. Without costs for filtration, their number rises to 63.

In the case of replacing groundwater with reclaimed urban wastewater (Hypothesis II), three plants meet the minimum PE threshold, two with a primary temperature of 20 °C and one with a capacity of 250,000 PE at 15 °C. Twenty-five are the WWTPs that discharge onto ground, meeting the size threshold, when the filtration process is included. In addition, in this case, 63 are the WWTPs for which tertiary treatment would be economically feasible when filtration processes are not applied.

The total economically feasible volume of reclaimed urban wastewater, under the best economic conditions (i.e., primary temperature of 20 °C and without filtration) would be 69,410,841 m$^3$ under Hypothesis I and 72,387,419 under Hypothesis II. Under the best economic conditions, there appears to be no difference between the two hypotheses for reclaimed wastewater reuse in agriculture. On the contrary, with a primary temperature of 15 °C, the total economically feasible volume decreases by 30% (21,380,278 m$^3$) in the case of new irrigated land (Hypothesis I) and by 12% in the case of

reclaimed urban wastewater intended to preserve groundwater from saltwater intrusion (Hypothesis II). Finally, the filtration process makes the biggest difference in volume. With filtration, the economically feasible volume of reclaimed wastewater decreases by 69% and 66%, respectively, under the first and second hypotheses.

The following methodological step was performed to determine the volume of readily available reclaimed wastewater. Table 7 shows the results as an example of the analysis which was carried out. The estimates refer to the best economic conditions only. This made it possible to assess maximum volumes.

**Table 7.** Reclaimed urban wastewater readily available for irrigation in Puglia (2015).

| | Current Effluent Discharge | Primary Temperature | Filtration | Collective Delivery Networks | Plants | Volume (m³/Year) |
|---|---|---|---|---|---|---|
| | Surface water | 20 °C | Yes | Capitanata | 1 | 5,502,778 |
| | | | | Terre d'Apulia | 1 | 15,877,500 |
| | | | | Capitanata | 2 | 3,244,108 |
| New irrigated land | | | | Terre d'Apulia | 9 | 11,495,047 |
| | Ground | 15 °C | No | Arneo | 11 | 6,416,634 |
| | | | | Stornara e Tara | 6 | 3,353,941 |
| | | | | Ugento Li Foggi | 8 | 6,468,910 |
| | | | | ARIF | 6 | 3,238,441 |
| | | | | Municipal networks | 5 | 1,594,570 |

As reported in Table 7, under the best economic conditions, the readily available volume amounts to 57,191,929 m³. In the case of reclaimed wastewater reuse aimed at preserving groundwater, the volume is slightly higher (59,722,670 million m³ per year).

The result of the analysis to determine the readily available volumes of reclaimed urban wastewater for irrigation is illustrated in Figure 3.

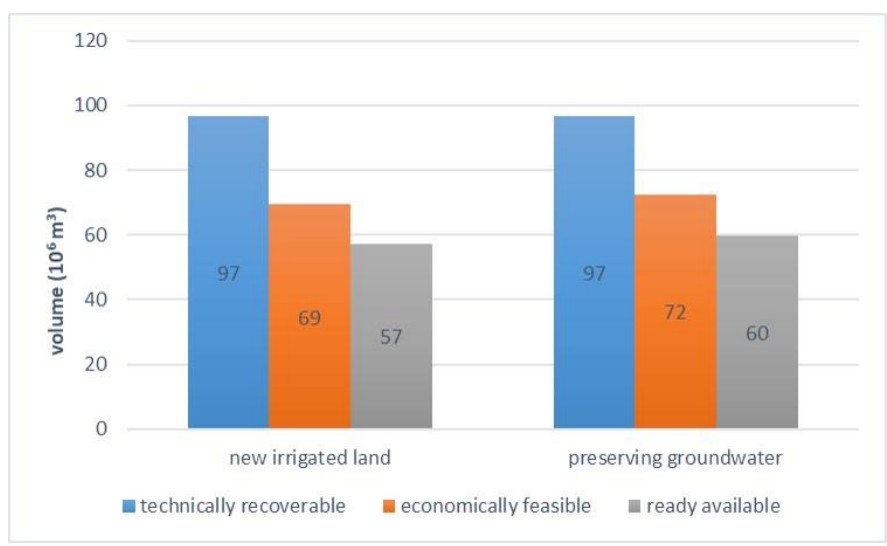

**Figure 3.** Technically recoverable, economically feasible and readily available reclaimed wastewater volumes.

As a whole, the volume of reclaimed urban wastewater for irrigation differs slightly under the two Hypotheses. If the reclaimed urban wastewater is intended for new irrigated land, there is a difference of almost 40 million m³ between the technically recoverable volume and the volume that is actually deliverable at the farm gate. Of that amount, 70% is not economically feasible and the remainder is outside the existing collective delivery networks.

The economically feasible and readily available volume of reclaimed urban wastewater within the groundwater vulnerable zone is reported in Table 8. With the exception of a small number of WWTPs in inland areas, the majority of economically feasible plants located within the collective delivery networks can be used to replace the groundwater source and so contrast saltwater intrusion along the coastline. Under Hypothesis II, less than 5 million m$^3$ out of a total of 60 million m$^3$ are outside the groundwater vulnerable zone. Nevertheless, within the Province of Foggia there is no chance of using the technically recoverable, economically feasible and readily available reclaimed urban wastewater volumes to contrast groundwater salt intrusion.

**Table 8.** Volume (m$^3$) of readily available reclaimed urban wastewater within the groundwater vulnerable zone (Hypothesis II).

| Province | Current Effluent Discharge | |
| --- | --- | --- |
| | **Surface Water** | **Ground** |
| Bari | 20,738,192 | 2,305,085 |
| Barletta-Trani-Andria | 3,651,711 | 2,522,823 |
| Brindisi | 2,904,476 | 8,789,333 |
| Foggia | 0 | 0 |
| Lecce | 6,013,258 | 1,566,739 |
| Taranto | 2,343,300 | 4,383,303 |
| **Sub total** | 35,650,937 | 19,567,282 |
| **Total** | 55,218,219 | |

## 4. Discussion and Future Research Issues

Water authority technicians and regional governments recommend reclaimed urban wastewater for reuse in agriculture as a solution for alleviating water scarcity in Mediterranean areas. Although reclaimed wastewater could potentially make up a significant proportion of irrigation water volume, it currently only accounts for a very small part. In this paper, we used the case study of Puglia (Italy) to highlight the main bottlenecks, namely technical and economic issues, which are hampering reuse of reclaimed urban wastewater in agriculture.

Our findings revealed that scaling down from potential to technically recoverable volume, the main barrier is the shortness of the irrigation season. Most irrigated crops in Mediterranean areas have a six-month irrigation period, as is the case in Puglia. On this basis, tertiary treatment systems of WWTPs can, at the most, run for half the year. Alternatively, reclaimed water can be stored in artificial wetlands [32] or aquifers to replenish groundwater resources. There are, however, some legal and administrative barriers to this in Italy. On the other hand, the results obtained with a primary temperature of 20 °C, which is common in the Mediterranean region during the spring and summer, revealed some technical advantages in reclamation treatment of wastewater for reuse. The irrigation season usually coincides with the hottest period of the year, thus there would be technical synergies. The latter result is merely theoretical therefore future research should try to conduct practical experiments in this regard.

Turning to economic issues, the most relevant aspect is connected to filtration when considering upgrading of WWTPs with effluent discharge into surface water. When considering additional costs of filtration, the economically feasible volume of reclaimed water is more than 60% lower than the technically feasible volume. This result differs from those of previous studies [5,22,33], in which the standards for reclaimed urban wastewater were pointed out as the most relevant economic concern. One of the main conclusions previously reached has been that the Italian legal framework does not provide for widespread wastewater reuse in agriculture [22]. In line with general opinion, limits for phosphorous and nitrogen have been relaxed and raised to 10 and 35 mg/L, respectively (Regional Regulation No. 8 of 18 April 2012 [34]). Moreover, Puglia's local government authorities recently reformed the legal framework so that the cost of tertiary treatment is now fully covered by domestic users, as they are the real polluters of fresh water. Despite the introduction of important reforms at

regional level, aimed at alleviating economic barriers to wastewater reuse in agriculture, the impact (if any) of the new framework is not yet clear. Based on our findings, the lack of reliable irrigation networks is also a very significant economic issue. Almost 30% of reclaimed water cannot be conveyed to the right place, namely the farm gate. This issue was also recognised in a recent report for pilot areas in the European Union [35]. Furthermore, supplying reclaimed wastewater involves additional costs for system monitoring and delivery management. The former refers to the regular monitoring of quality parameters and the latter to the additional costs (e.g., infrastructure and operating costs) of handling steady flows from the WWTPs and meeting irregular irrigation demands. Both issues are strictly related to a maximum 24-h storage period established by national authorities. In this regard, more research is needed in the field of innovative monitoring systems capable of facilitating continuous and instantaneous quality checks and recording performance data. Lastly, as reported in [21], although farmers' acceptance of reclaimed urban wastewater does not seem to be a problem, they would only use such a source at a lower price (tariff) than current conventional irrigation water sources.

Generally speaking, and in common with many other assessments of reclaimed urban wastewater for reuse in agriculture, the technical and economic data used in this study refer to average estimates. This is because there are still few WWTPs with tertiary systems that have been operating for a long time. In the future, more empirical analyses should be carried out using actual data, with the aim of also verifying and adjusting estimates made in previous studies. Moreover, we ran our estimates taking into account the technological state of art in Puglia (e.g., gravity filtration), while on new plants, the entire treatment architecture could be modified to include, e.g., membrane filtration. In this regard, membrane filtration is not yet implemented and in such case estimates will be based on pilot studies [36]. Finally, all the estimates used in this study as well as in previous analyses are based on assumptions that do not consider transaction costs. In practice, wastewater treatment for reuse and irrigation water delivery services are usually provided by different economic entities (either private firms or public utilities). Agreements therefore have to be drawn up to establish the obligations of the parties concerned, which leads to transaction costs. Although this topic has never been previously addressed, it is very relevant, especially in the early stages of wastewater reuse in agriculture, as is the case in Puglia.

## 5. Concluding Remarks

In this study, the case of Puglia (Italy) was investigated. Assessment of reclaimed urban wastewater volume for reuse in agriculture was carried out and the results were discussed in terms of the implications for policy-making and pointing out future research needs. The findings showed that the main technical barrier for reuse in agriculture is the shortness of the irrigation season. On the other hand, the main economic concern is related to filtration followed by lack of conveyance systems. When considering additional costs of filtration, the economically feasible volume of reclaimed wastewater is more than 60% lower than the technically feasible volume. This result differs from those of previous researches in which attention has been drawn on the Italian legal framework and the quality parameters for reuse of reclaimed urban wastewater. Further research should also address the issue of transaction costs by establishing the obligations of wastewater treatment plants to deliver reclaimed wastewater services to farmers.

It is worth a mention that our results are based on estimates, like most published work on this topic. This is usually the case for several Mediterranean regions where there are still few WWTPs with tertiary treatment that have been operating for a long time. Future research should try to include assessment based on actual data.

**Author Contributions:** S.A., V.I. and A.F.P. conceived the technical sequences of WWTP and assessed polishing costs; G.G. and B.C.d.G. conceived the framework analysis and ran economic study. G.G. wrote Sections 1, 2.1.2 and 3.2, S.A. wrote Section 3.1, B.C.d.G. wrote Section 4, V.I. wrote Sections 2.1 and 2.1.1. All authors had an equivalent contribution to writing Section 5.

**Funding:** This research is carried out within the project "Economia delle risorse irrigue in Puglia" CUP B37G17000030007, funded by Regional Government of Apulia. Funds do not cover the costs to publish in open access. This work was also realised within "Re-water" project which is co-founded by European Union, European Regional Development Funds (E.R.D.F.) and by National Funds of Greece and Italy. Furthermore, the present investigation was carried out with the support of Apulian Region (POR Puglia FESR-FSE 2014-2020) by means of the project entitled "T.E.S.A.".

**Acknowledgments:** We greatly appreciate all helpful suggestions and comments from anonymous reviewers. Special reference is for Editor.

**Conflicts of Interest:** The authors declare no conflict of interest.

## Appendix A

| Parameter | Unit of Measure | Surface Water Discharge | Ground Surface Discharge | Reuse in Agriculture |
|---|---|---|---|---|
| pH | - | - | 6–8 | 6–9.5 |
| SAR | - | - | 10 | 10 |
| Total Suspended Solids | mg/L | ≤35 | 25 | 10 |
| BOD5 | mg $O_2$/L | ≤25 | 25 | 20 |
| COD | mg $O_2$/L | ≤125 | 100 | 100 |
| total phosphorus | mg P/L | - | 2 | 2 * (10) |
| total nitrogen | mg N/L | - | 15 | 15 * (35) |
| pathogens (*Escherichia coli*) | UFC/100 mL | - | <5.000 | 10 |

* Puglia Regulation No. 8 of 18 April 2012.

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
