# Peer review of "Assessing Reclaimed Urban Wastewater for Reuse in Agriculture: Technical and Economic Concerns for Mediterranean Regions"

_water, doi:10.3390/w11071511_

Round 1

Reviewer 1 Report

The paper is important since it describes a growing problem of water deficit in agricultural production and which can partly be solved by use of domestic wastewater.

However, the paper must be improved before publication. One thing of concern are the technical descriptions which are confusing or unclear. What is meant with filtration? Maybe prepare a table showing which treatment technologies the WWPTs have. 

Conclusions should focus on the cost-benefit analysis you have in the objectives and its results.

In supplementary material is a map without legend and scale, what it is for?

See my remarks in the appended manuscript.

Author Response

We appreciate very much all suggestions received by reviewer. We carefully took into consideration all remarks into the text. A flow chart has been enclosed in the main text, where steps are better understandable. Supplementary material has been dropped out. The main text has been also revised, in order to explain deeply the methodology, results and discussion. Some English flaws has been corrected. Terminology on wastewater much more accurate.

Some other answers to comments are as follow:

What is meant with filtration?

Filtration is a mechanical process of separating suspended and colloidal particles from water. Usually it follows coagulation, flocculation, and sedimentation.

In order to remove particles water flows through a medium made of different layers of sand and gravel, sometimes, anthracite.

Filters in general can work either with upward or downward flowing water. In the second case the process can be driven under pressure or by gravity. Pressure bed filters are often called rapid filters and may require more pushed preliminary treatments. Gravity units are slower processes but some economical advantage in operation and management.

As filtration is the main item for economic feasibility, reader can find much more details on the cost in the paper by Arborea et al. 2017 with related supplementary materials.

Line 111: Looking at your hypotheses, it seems that water as such is most important, not the added value of nutrients in the wastewater. When reading this section it is unclear to me why you spend much focus on the "productive factor" caused by nutrients and not water to avoid effects of droughts.

The reclaimed wastewater might have additional value derived from nutrients such as N and P. Although already recognised by some authors (Vivaldi et al., 2017; Vergine et al., 2017) there are still several unsolved issues. Actually, reduction in the use of fertilisers can be achieved only with appropriate crops management. This refers to the fact that each crop should have appropriate management. We took an average reference of crops management under the current status quo (i.e. business as usual). Moreover, the economic value of irrigation water (regardless of source) increases during drought period and decreases when water become abundant. The irrigation water value of reclaimed wastewater should be assessed as average value during the life cycle of wastewater treatment plant, namely 25 years.

Line 336: This is not a technical barrier! Water is simply needed only during the vegetation/cropping period.

May perhaps be. In any case, we assessed the technically recoverable volume for irrigation (Table 5). If the reclaimed wastewater is for other reuse, technically recoverable volume will also change. 

Reviewer 2 Report

Specific comments are made in the appended manuscript, so that the authors can further improve the manuscript.

Author Response

Dear reviewer we recognise that in the previous version some English flaws and misleading terminology on wastewater could lead reader to confusing results. We appreciate very much all suggestions and comments received. We did our best in order to improve the text and terminology throughout the manuscript as well as description of analysis conducted. Moreover, the chart flow reporting the steps of analysis is shown. We offer a revised version of paper in which we tried to improve previous version on basis of comments received by three anonimus experts. All change made are traced in the appended manuscript.

There are also other highlighted yellow sentences without specific comment. In this case, we try to give some explanation on the basis of our understanding, as follow:

Line34 direct reuse: There are two type of wastewater reuse: direct and indirect. In direct reuse reclaimed wastewater is piped into some type of water systems for irrigation without first being diluted in a natural stream such as groundwater. Indirect reuse involves mixing of reclaimed wastewater with another body of water before reuse. Example is the recharge of groundwater aquifer by reclaimed wastewater. We focused on direct reuse being (within Italian legal framework) groundwater recharge with reclaimed wastewater forbidden.

Note at table 1 was firstly required by other reviewer. In this way, we try to make self-explaining the table.

Line 163-164: same as Line 145-147?

To our understanding this sentence is not replicate any more.

Line 199: Why 92? Where is another one?

We already reported in the note below the table that data of wwtp in Uggiano la Chiesa was not available. We have rephrased the sentence.

Table 8: why zero for Foggia?

The Province of Foggia reported zero amount of reclaimed wastewater for reuse in agriculture intended to groundwater replenishment. Indeed, there is none WWTP which fits into the selection criteria of our analysis. Namely, the collective irrigation networks do not cover the groundwater vulnerable zone.

Line 283: the main barrier is the shortness of irrigation season. There was unclear terminologies and poor description of technically recoverable volume. In fact, we consider the potential available volume and the technically recoverable volume. The potential volume refers to effluent amount of wastewater flowing through the WWTPs over a year. Such volume can be potentially treated to convert it into reclaimed wastewater.  The technically recoverable for irrigation is the fraction of potentially treated that can be effectively reused in agriculture. Thus it depends also on the irrigation demand. Improvement on this issue are now provided.   

Line 287 What parameters. This sentence has been deleted. We focus the discussion on cost-benefit results.

We offer an example of calculation.

We obtained the annual recoverable volume by multiplying the daily water availability of WWTPs. We assigned an average value for the volume of reclaimed wastewater according to the plant capacity expressed in PE as in the following table.

Plant Capacity

(PE)

Water Availability

(m3/day)

2,000

432

5,000

1,080

10,000

2,160

20,000

5,760

30,000

8,640

40,000

11,520

50,000

14,400

70,000

20,160

100,000

28,800

250,000

90,000

500,000

180,000

In the Appendix, PH and other parameters without values do not require monitoring. 

Reviewer 3 Report

It is a pleasure for me to review a paper about assessing wastewater reuse in agriculture. The paper is well written and I do not have any major remarks for modifications. 

In the "materials and methods" section, you must do a framework figure to be better to visualize  the steps you have used for the analysis. 

In the figure 1 - make the map at 300 DPI resolution, put the neighbors of the study area, the map scale, the north and some information (name of cities) to to fit the study area.

Instead of chart 1 from line 256 , write figure 1. Remove the green background from this figure and add an explanation for the figure. 

Author Response

Dear reviewer, we are very grateful for your comments. A flow chart has been enclosed in the main text, where steps are better understandable. Figure 1 is already at highest resolution. Map details has been added as requested. Chart 1 is now renamed Figure 2. Green background was not intended. We have tried to avoid it.

The main text has been also revised, in order to explain deeply the methodology, results and discussion. Some English flaws has been corrected. Terminology on wastewater much more accurate.

Reviewer 4 Report

The aim of this work is to improve the assessment of wastewater reuse in agriculture. In particular, it attempts to shed light on the technical and economic challenges for Mediterranean areas. The region of Puglia (in south-eastern Italy) was used as a case study to assess wastewater reuse for irrigation. A spatial cost-benefit analysis was conducted in order to quantify and locate the volume of technically and economically feasible and readily available treated wastewater. The results of the cost-benefit analysis were discussed, considering their implications for policy-making and pointing out future research needs.

The work is interest, because in the face of global climate change, there is a need for a sustainable management of water resources, which includes water conservation, and where the water reuse is an important strategic component.

The paper had significant changes regarding the first revision. Therefore, I consider that que present work is suitable for publication.

Author Response

Dear reviewer, we are very grateful for your comments. 

Further improvent has been made. The main text has been also revised, in order to explain deeply the methodology, results and discussion. Some English flaws has been corrected. Terminology on wastewater much more accurate.

Round 2

Reviewer 2 Report

L 102: I believe, sections 2.2 and 2.3 should be under section 2.1; rewrite as 2.1.1 and 2.1.2, respectively.

L278: Rephrase or delete the whole paragraph. The information is already given in the "Methodology" section.

L321: Please correct, there is no word called "annex 1"

Appendix: Please include the sample calculation as suggested in rebuttal.

Author Response

Thanks again for all assistance.

We appreciate very much all suggestions. Sections 2.2 and 2.3 have been listed as suggested.

Line 278 deleted.

L321 corrected.

In Appendix A the sample calculation as suggested in rebuttal enclosed.

Other minor editing flaws also revised.

This manuscript is a resubmission of an earlier submission. The following is a list of the peer review reports and author responses from that submission.

Round 1

Reviewer 1 Report

Aim of the study is to improve an assessment of wastewater reuse in agriculture in Mediterranean areas, with focus on  Puglia as a case study. Spatial cost-benefit analysis is carried out in order to quantify and locate technically feasible, economically convenient and readily available treated wastewater volume.

The paper is written in poor "english" (perhaps using Google translator?) and should be thoroughly revised from the grammar and logic points of view. The style is redundant, repetitive  and poorly apt to convey a clear message.

Perhaps, as the authors state "our results are merely theoretical, but future research should try to conduct practical experiments in this regard".  In fact, the criteria upon which economic feasibility of wastewater reuse are evaluated are not clear at all.

Authors state that " The economic feasibility of wastewater reuse in agriculture stems from the methodological framework as implemented by Arborea et al. [10] in Puglia. Only the real economic benefits of reclaimed water as a productive factor for irrigation are taken into account."

It would seem that the main discriminant for economic feasibility is treatment costs, however, what about actual irrigation NEEDS and CROP VALUE? 

Actual WWT technology seems not to have an impact on the assessment, while it is clear that it does. at one point it is stated that "....  the most relevant aspect is filtration. With filtration the

economically feasible volume of wastewater is more than 60% lower than the technically feasible

volume". As exposed, the concept is quite obscure, perhaps a more detailed explanation of this statement is due. I am not even sure of WHAT kind of filtration we are talking about at this point. 

"The main conclusion has been that the Italian legal framework refrains from widespread wastewater reuse in agriculture". This statement is probably true, but did we need this long manuscript to  prove it?

In summary, the paper summarizes some data (existing) adding some unconnected, almost random considerations, to arrive at the conclusion that "..., as in many other researches where the assessment of wastewater reuse in agriculture has been carried out, in this study technical and economic data refer to quite theoretical average figures. This is because there are still few long-standing WWTPs. In the near future, empirical analyses with actual data should be increased also with the aim of checking and adjusting estimates made in previous studies."

Which is not a conclusion at all.

Author Response

Reviewer 1

Aim of the study is to improve an assessment of wastewater reuse in agriculture in Mediterranean areas, with focus on  Puglia as a case study. Spatial cost-benefit analysis is carried out in order to quantify and locate technically feasible, economically convenient and readily available treated wastewater volume.

The paper is written in poor "english" (perhaps using Google translator?) and should be thoroughly revised from the grammar and logic points of view. The style is redundant, repetitive and poorly apt to convey a clear message.

Dear Reviewer we appreciate very much the effort made and we took all comments as best chance to get better our work. On the English Editing, we apologise this inconvenient. Before any submission, English revision is carried out by external services. This time may be the topic of wastewater was not within the skills of such services. The English revision has been carried out again. We hope this time will be apt. On the style, quite redundant, actually there are no clear suggestions from your point of view. However, we did our best in order to give a clear message. A point-to-point reply is provided as follow.

Perhaps, as the authors state "our results are merely theoretical, but future research should try to conduct practical experiments in this regard".  In fact, the criteria upon which economic feasibility of wastewater reuse are evaluated are not clear at all.

Authors state that " The economic feasibility of wastewater reuse in agriculture stems from the methodological framework as implemented by Arborea et al. [10] in Puglia. Only the real economic benefits of reclaimed water as a productive factor for irrigation are taken into account."

It would seem that the main discriminant for economic feasibility is treatment costs, however, what about actual irrigation NEEDS and CROP VALUE? 

All previous comments make reference to economic assessment of wastewater reuse in agriculture. As reported in the text, we took advantages of previous research by Arborea et al. [10]. They have recently carried out a cost-benefit analysis in Puglia starting from irrigation requirement and crop values. In Giannoccaro et al [26], monetary value of irrigation water in Puglia is reported taking as irrigation volume the region’s average value of 2,475 m3/ha. The economic benefit of preserving groundwater from salt intrusion in Puglia is from Giannoccaro et al [27]. We used all this information to assess the economic feasibility. Accordingly, the threshold value as reported in table 1 has been elicited. Likely, for unskilled readers this way to establish economic feasibility can appear unconventional. In order to improve our text, additional details on methodological process as well as sources used have been enclosed.

Our results are merely theoretical as most of already published researches on this topic. This is usually the case of several Mediterranean regions were wastewater reuse is at the early stage of development. By the way, the theoretical nature refers to the fact that real data on the advantages of primary temperature at 20° C and related reduction in the cost of wastewater treatment in Puglia and wherever are not yet available. We would like to stress that apart of the assessment, the research aim is to support policymakers in their decisions and at the same time to point out future research needs as stated at the beginning “The Puglia (Italy) case is used as an example and results are discussed with the aim of supporting policymakers’ decisions and pointing out future research needs.” as well as in the conclusion section “In this research the case study of Puglia (Italy) has been used with the aim of highlighting the main bottlenecks, namely technical and economic, which are hampering wastewater reuse in agriculture.”

In any case, by an in-depth English editing this message is made clearer. 

Actual WWT technology seems not to have an impact on the assessment, while it is clear that it does. at one point it is stated that "....  the most relevant aspect is filtration. With filtration the economically feasible volume of wastewater is more than 60% lower than the technically feasible volume". As exposed, the concept is quite obscure, perhaps a more detailed explanation of this statement is due. I am not even sure of WHAT kind of filtration we are talking about at this point. 

We recognize the relevance of this point as raised by the reviewer while we recognize that such conclusion is obscure without considering elements that were actually provided in Arborea et al. [10]. In the  first submission manuscript we omitted those details for the sake of brevity but, according to the reviewer’s suggestion we have now introduced more explanations (see lines 127-136 section 2.2) and revised related sentences in abstract and conclusions. A specific description of what kind of filtration has been considered has been also added ((see lines 137-141 section 2.2).

"The main conclusion has been that the Italian legal framework refrains from widespread wastewater reuse in agriculture". This statement is probably true, but did we need this long manuscript to prove it?

Ok, this sentence refers to previous works. There was a misunderstanding, may be arose from the quality of English editing. We have re-edited the text.

In summary, the paper summarizes some data (existing) adding some unconnected, almost random considerations, to arrive at the conclusion that "..., as in many other researches where the assessment of wastewater reuse in agriculture has been carried out, in this study technical and economic data refer to quite theoretical average figures. This is because there are still few long-standing WWTPs. In the near future, empirical analyses with actual data should be increased also with the aim of checking and adjusting estimates made in previous studies."

Which is not a conclusion at all.

This is one of main conclusion we pointed out and it is true. Reviewer is specially focusing on this. We stated that our elaboration as well as others we mention in literature use estimates as main data source. Real data of WWTPs in Apulia Region are not yet available because the longest-operating WWTP for reuse started in 2012. Moreover, average means that estimates of operational costs refer to fully operative plant. This assumption is totally theoretical being still now unreached in Apulia WWTPs.   

By the way, we stated that filtration and shortness of irrigation season make the economy of wastewater reuse in agriculture unprofitable. Moreover, making agreements between WWTP and irrigation services, where liabilities are established leads to transaction costs.  Despite being very relevant, especially in the early stage of wastewater reuse in agriculture – as is the case for Puglia -, this topic has never been tackled before. On the technical concerns, we explore potential effects of higher temperature of primary sedimentation. Finally, advance in systems monitoring in order to provide WWTPs for reuse with on-time and cheap out-flow quality check is envisaged. 

As suggested by other reviewer, a section of concluding remarks has been added.

Reviewer 2 Report

The aim of this work is to improve the assessment of wastewater reuse in agriculture. A spatial cost-benefit analysis was carried out in order to quantify and locate the technically feasible, economically convenient and readily available treated wastewater volume. The Puglia (a region in South-eastern Italy) case was used as an example and results was discussed with the aim of supporting policymakers’ decisions and pointing out future research needs.

The work is interest, because in the face of global climate change, there is a need for a sustainable management of water resources, which includes water conservation, and where the water reuse is an important strategic component. Therefore, I consider that que present work is suitable for publication after minor modifications:

I suggest that the word “reuse” be inserted in the “Keywords”.

I suggest that the section 4 be only “Discussion” and that a section 5 of “Conclusions” be inserted. So the lines 84 and 85 should be reviewed.

In Figure 1 and Chart 1, the legend should be in the bottom.

The Italian laws – D.L. 152/06, D.M. 185/2003 and the Regional Regulation of 18 April 2012, no. 8, cited in the text, should be included in the “References”.

In the Table 2, columns 3 and 4, where is 975 and 946, should be 0.975 and 0.946, respectively. This table is not mentioned in the text. You must mention it.

In the Table 5, column 7, where is 7,637454, should be 7,637,454. In this table, check the rounding of the values in the “Total”.

Author Response

Reviewer 2

Comments and Suggestions for Authors

The aim of this work is to improve the assessment of wastewater reuse in agriculture. A spatial cost-benefit analysis was carried out in order to quantify and locate the technically feasible, economically convenient and readily available treated wastewater volume. The Puglia (a region in South-eastern Italy) case was used as an example and results was discussed with the aim of supporting policymakers’ decisions and pointing out future research needs.

The work is interest, because in the face of global climate change, there is a need for a sustainable management of water resources, which includes water conservation, and where the water reuse is an important strategic component. Therefore, I consider that que present work is suitable for publication after minor modifications:

I suggest that the word “reuse” be inserted in the “Keywords”.

Ok, done.

I suggest that the section 4 be only “Discussion” and that a section 5 of “Conclusions” be inserted. So the lines 84 and 85 should be reviewed.

 Ok, thank you.

In Figure 1 and Chart 1, the legend should be in the bottom.

 Ok, done

The Italian laws – D.L. 152/06, D.M. 185/2003 and the Regional Regulation of 18 April 2012, no. 8, cited in the text, should be included in the “References”.

Ok, done. 

In the Table 2, columns 3 and 4, where is 975 and 946, should be 0.975 and 0.946, respectively. This table is not mentioned in the text. You must mention it.

 Ok, done.

In the Table 5, column 7, where is 7,637454, should be 7,637,454. In this table, check the rounding of the values in the “Total”.

Many thanks.

Round 2

Reviewer 1 Report

The manuscript has not been substantially improved from its previous version.

There is  a  huge source of confusion in the manuscript introduced in Section 2.3 where the authors refer to Tab.1 and Tab 4. OF  ITALIAN DISCHARGE STANDARDS. In the subsequent text (line 128 on) these can easily be confused with Tables 1 and 4 of the manuscript. Perhaps the introduction of a NEW Table 1 indicating the prescriptions of Tab 1 and 4 of the standards (which are never explicitly indicated) could solve this. 

Also, I am wondering what the authors mean with FILTRATION: there are many types of filtration, achievable with different technologies. I assume that on existing plants this would be probably sand filtration, but on new plants the entire treatment architecture could be modified to include, i.e. membrane filtration in the process. This would change the entire parametrization of treatment costs, and statements like those in lines 137-141 would have to be re-evaluated. The authors never discuss wastewater treatment technology, and this seems to be an equally important issue to consider, other than delivery of water. Also, the authors mention nitrification/denitrification efficiency issues: if water is used for irrigation, this process should probably be eliminated (providing N to crops) with dual financial advantage (lower treatment costs, lower fertilizer-N supply). 

Figure 1 shows gwater vulnerability in Apulia: given the dramatic situation of salt intrusion and impending salt intrusion (indicated as Q&Q worsening) would it not make sense to adopt everywhere a policy of treated wastewater infiltration? In the long run this could possible improve GW conditions at large and irrigation indirectly.  It would be interested to see (e.g. with dots on the map) WHERE the WWTPs are located. 

The statement in lines 283-4 is nonsense: perhaps it is meant that WTTP effluent can only be used (for irrigation) for only half a year. Still it could be used for aquifer recharge the rest of the time, which would be still beneficial, given the general situation.

I cannot but re-confirm my previous assessment of the manuscript, i.e., rejection.